# Atrial Fibrillation Prediction Based on Recurrence Plot and ResNet

**DOI:** 10.3390/s24154978

**Published:** 2024-08-01

**Authors:** Haihang Zhu, Nan Jiang, Shudong Xia, Jijun Tong

**Affiliations:** 1School of Information Science and Engineering, Zhejiang Sci-Tech University, Hangzhou 310018, China; haihangzhu_zstu@163.com (H.Z.); jiangn@zstu.edu.cn (N.J.); 2The Fourth Affiliated Hospital Zhejiang University School of Medicine, Jinhua 321000, China; shystone@zju.edu.cn

**Keywords:** atrial fibrillation, ECG, prediction, Recurrence Plot, ResNet

## Abstract

Atrial fibrillation (AF) is the most prevalent form of arrhythmia, with a rising incidence and prevalence worldwide, posing significant implications for public health. In this paper, we introduce an approach that combines the Recurrence Plot (RP) technique and the ResNet architecture to predict AF. Our method involves three main steps: using wavelet filtering to remove noise interference; generating RPs through phase space reconstruction; and employing a multi-level chained residual network for AF prediction. To validate our approach, we established a comprehensive database consisting of electrocardiogram (ECG) recordings from 1008 AF patients and 48,292 Non-AF patients, with a total of 2067 and 93,129 ECGs, respectively. The experimental results demonstrated high levels of prediction precision (90.5%), recall (89.1%), F1 score (89.8%), accuracy (93.4%), and AUC (96%) on our dataset. Moreover, when tested on a publicly available AF dataset (AFPDB), our method achieved even higher prediction precision (94.8%), recall (99.4%), F1 score (97.0%), accuracy (97.0%), and AUC (99.7%). These findings suggest that our proposed method can effectively extract subtle information from ECG signals, leading to highly accurate AF predictions.

## 1. Introduction

Atrial fibrillation (AF) is characterized by rapid and irregular ectopic atrial electrical activation, on the basis of mostly reentrant but also focal mechanisms [1,2]. Mechanically, this leads to irregular and inefficient atrial and ventricular contractions that may have considerable detrimental effects on the cardiac output. It can lead to heart failure, but also to thromboemolic complications like stroke, caused by thrombus formation in the fibrillating atria. To prevent such complications, patients with AF are usually treated with anti-coagulants. Oftentimes, AF slowly becomes manifest, starting with short episodes of paroxysmal AF. Early detection of AF, even if it does not yet have much hemodynamic impact is, hence, very important for the timely institution of anti-coagulatory therapy. Data from the Framingham Heart Study (FHS) indicate that AF affects approximately 10 million individuals in Europe and 2.3 million people in the United States [3]. In China, approximately 4.87 million individuals are living with AF. The prevalence of AF is positively correlated with age; the older the age, the higher the prevalence. Among those over 80 years old, more than 10% have AF [4,5]. Given its significant health impact, developing an accurate AF prediction model is crucial. Traditional methods primarily rely on manual analysis of ECG signal waveforms, including P-waves, QRS complexes, and T-waves, to detect AF. However, AF often exhibits sudden, intermittent, and short-term characteristics in its early stages, making early detection challenging. Delayed monitoring can provide a more comprehensive diagnosis but is expensive and can lead to waste of medical resources when implemented on a large scale. In clinical practice, many patients with suspected AF experience recurrent chest tightness and palpitations, but their ECGs may show no abnormalities at the time. Through delayed cardiac monitoring, 6–40% of AF cases may be overlooked [6]. Therefore, accurate and effective prediction of AF is crucial.

Recently, with the growing demand for portable cardiac monitoring devices, the diagnosis of AF based on single-lead ECG signals [7] has become a research hotspot. Low-cost, portable monitoring systems are expected to be achieved, offering a more accessible and convenient means of detection. Currently, research tasks [8] in the field mainly fall into two categories. The first is detection, which involves identifying AF patients from all the participants. The second is prediction, where, based on the ECG data collected before AF episodes, the goal is to predict whether AF events will occur in the future, thereby safeguarding the patients’ life and well-being. In these tasks, the main approach used for analyzing ECG signals is machine learning and deep learning [9,10,11,12,13]. For instance, Boon et al. [14] extracted time-domain, frequency-domain, non-linear, and bispectral features from quantified heart rate variability (HRV), and achieved a prediction accuracy of 79.3% based on the classifier of the Support Vector Machine (SVM). Rouhi et al. [15] utilized the SHapley Additive exPlanations (SHAP) technique and Random Forest (RF) for the classification of ECG signals in AF detection, and the average F1 score was 74.6%. Bashar et al. [16] proposed a novel machine learning method based on Density Poincaré Maps. K-Nearest Neighbor (KNN), SVM, and RF classifiers were employed for the classification of AF, premature atrial contraction (PAC), and premature ventricular contraction (PVC). The SVM exhibited the best performance, achieving an accuracy of 98.99%; this approach resulted in further improvement in classification accuracy. Liaqat et al. [17] employed various machine learning classifiers, including SVM, logistic regression, Multilayer Perceptron (MLP), and eXtreme Gradient Boosting (XGBoost), for AF detection; the accuracy achieved by each classifier was 71.2%, 70.8%, 65.7%, and 68.2%, respectively. From the above-mentioned machine learning methods, it can be observed that entropy measurement, HRV analysis, algorithm selection, etc., have played significant roles in AF detection or prediction. AF detection has achieved high accuracy, but there are limitations in accuracy for prediction tasks [18,19]. Additionally, manual feature extraction is challenging as it cannot encompass all the hidden characteristics. ECG signals are susceptible to disturbances such as baseline drift, etc. [20], which can have a considerable impact on the classification results.

Given these considerations, some deep learning methods have been employed to address the aforementioned challenges. Aschbacher et al. [21] utilized HRV measurements and Long Short-Term Memory (LSTM) neural networks to differentiate between AF and sinus rhythm, achieving an AUC of 95.4% and a sensitivity of 81.0%. Serhal et al. [22] utilized wavelet decomposition to identify features predictive of AF, achieving a prediction accuracy of 94.0%. Cai Wu et al. [23] simultaneously extracted P-wave morphological parameters and HRV feature parameters from ECG and constructed a deep learning model, obtaining an accuracy of 92.0% and a sensitivity of 88.0%. Le Sun et al. [24] employed a recursive neural network composed of stacked LSTMs for AF prediction, which better captures relevant features, achieving an accuracy of 92.0% and an F1 score of 92.0%. Shikha et al. [25] utilized ResRNN to classify arrhythmia disease (including AF), achieving a normal accuracy of 91.0%. Zhang et al. [26] proposed a novel Time-adaptive dense network named MP-DLNet-F, and solved the problem of incompatibility between variable-duration ECG and 1D-CNN, achieving a classification accuracy of 87.98%.

Overall, utilizing deep learning methods can effectively reduce the workload of manual feature extraction and enhance the model’s ability to handle noise interference. However, there are still some challenges. The mentioned methods achieve high accuracy, but the recall is low, indicating a problem of not detecting positive samples [27]. Additionally, ECG waveforms exhibit various morphologies, and during AF episodes, the abnormal waveforms differ from one another. This leads to the poor generalization ability of existing deep learning models [28].

Based on the above research, this paper proposes a method combining RP [29] and ResNet for AF prediction. Utilizing the features of RP can help uncover potential information related to AF episodes, while the ResNet network excels at recognizing varying degrees of chaos among different types of AF signals. Combining RP with ResNet maximizes the extraction of hidden information from ECG signals, enhancing the performance of AF prediction. This approach effectively addresses the time-consuming and labor-intensive issue of manual ECG signal analysis, thereby reducing the risk of missed diagnoses.

## 2. Database and Methods

### 2.1. Database

This study includes a total of 474,288 initial routine 12-lead ECG data collected from November 2014 to July 2022 at the Fourth Affiliated Hospital of Zhejiang University School of Medicine. All 12-lead ECGs were excluded under the following conditions: (1) Lack of diagnosis/incorrect diagnostic content (echocardiography, pulmonary function diagnosis conclusions); (2) Pacemaker rhythm; (3) Atrial flutter (the diagnosis includes “atrial flutter” but not “AF”); (4) Electrode misplacement, dextrocardia; (5) Atropine test; (6) Death ECGs; (7) Age less than 18 years. After the initial exclusion criteria, a total of 448,374 ECGs remained. The routine 12-lead ECGs screening and grouping process diagram is shown in Figure 1.

Based on whether patients experienced AF during the monitoring period, they were divided into two groups: the Pre-AF group and the Non-AF group. The Pre-AF group included ECGs from patients who had episodes of AF during the monitoring period (as recorded by the hospital’s routine 12-lead ECG and/or ambulatory ECG capturing AF episodes), and excluded ECGs without AF episodes within the subsequent 365 days and those after radiofrequency ablation surgery. The Non-AF group included ECGs from patients with no history of AF (no record of AF in the hospital’s ECGs, ambulatory ECGs, outpatient diagnoses, or past medical history), and excluded data from patients who were not followed up for a full 365 days. The group selection process is depicted in Figure 2.

After screening, the Pre-AF group in this database includes 1008 AF patients with 2067 ECG records, consisting of 2067 records from 365 days prior to the onset of AF (including 660 records from the 30 days before the onset). Among the Pre-AF group, 64.1% are male patients. The Non-AF group comprises 48,292 patients with 93,129 ECG records. Of these, 52.1% are male patients. General information for both groups is summarized in Table 1. The ECG image data contain 12 leads, with a recording time of 10 s for each lead at a sampling frequency of 500 Hz, stored in (xml) format. All patients provided informed consent prior to data collection.

The ECG data from the 30 days before the AF episodes showed better predictive results. Hence, 660 I-lead ECG records were selected, along with 1320 randomly chosen normal samples, following a ratio of 7:2:1 for model training, validating, and testing. Consequently, the final dataset was divided into a training set with 1386 samples, a validation set with 264 samples, and a test set with 132 samples. The data samples are labeled, where “1” represents the occurrence of AF events after 30 days, and “0” indicates that no AF is detected after 30 days.

The PAF Prediction Challenge Database (AFPDB) [30] used in this paper includes the following datasets:

(1) One hundred healthy group ECG records with a length of 30 min, named n01-n50 and n01c-n50c.

(2) ECG record sets from 25 PAF patients, where each record consists of two 30-min segments and two 5-min segments. The first 30-min segment represents the ECG record prior to the occurrence of PAF (named p01, p03…p49), followed by a 5-min segment capturing the duration of the AF episode (named p01c, p03c…p49c). The second 30-min segment represents a record taken at least 45 min before any AF event (named p02, p04…p50), followed by a 5-min segment capturing the duration of the AF episode (named p02c, p04c…p50c).

(3) All the records in the dataset have an initial sampling frequency of 128 Hz and a resolution of 12 bits. The relationship between different types of records is shown in Figure 3. “Non-AF” indicates ECG segments with no AF events during that period. “Fragment 45 min before onset of AF” refers to ECG segments from 45 min before the onset of AF, while “Fragment 30 min before onset of AF” refers to ECG segments from within 30 min before the onset of AF. “AF occurrence” represents the period when AF is occurring. Analyzing ECG segments from different time intervals relative to the onset of AF helps us identify and understand the various features of AF and its potential physiological and pathological mechanisms.

During the AF prediction task, 100 Non-AF records and 50 records from before AF episodes are used. To avoid having the same patient’s records in both the training and testing datasets, the 100 Non-AF records and 50 records from before AF episodes are randomly split into a ratio of 7:2:1 for model training, validating, and testing. Subsequently, the records are resampled to 500 Hz and segmented into 10-s fragments. The final dataset is divided into a training set with 1575 samples, a validation set with 450 samples, and a test set with 225 samples.

### 2.2. Method

The data processing flowchart is shown in Figure 4. It involves data preparation and data input, slicing ECG signals of different durations into 10-s segments, and normalizing them to address differences between databases. The ECG data undergo wavelet decomposition, threshold filtering, and reconstruction to remove noise interference. Subsequently, the 1-D time series is transformed into a 2-D recursive matrix before being fed into the network for prediction.

#### 2.2.1. Data Normalization

This paper employs the Min−Max normalization method [31] to process ECG signals with different amplitude distributions, mapping them to a unified range to facilitate feature extraction and prediction of ECG signals. For a given sample dataset X={x1,x2,…,xn}, where the minimum value of the sample is min and the maximum value is max, $x_{i}$ represents the feature value of the i-th sample. The normalized result of the sample’s feature value can be calculated using Equation (1), where ${x_{i}}^{\prime }$ represents the normalized feature value, which falls within the range [0, 1].
(1)$${x_{i}}^{\prime }=\frac{x_{i}-min}{max-min}$$

#### 2.2.2. Wavelet Decomposition and Reconstruction

The original ECG signals contain noise, primarily including baseline drift, powerline interference, and muscle artifacts [32]. Baseline drift is a low-frequency artifact mainly caused by factors such as respiration, poor electrode contact, and skin-electrode impedance. It appears as a slow-changing sinusoidal curve, as shown in Figure 5(A4). Muscle artifacts are caused by the electrical activity of muscles during contraction, for example, involuntary tremors in subjects. This noise appears as irregular, rapidly changing waveforms, predominantly high-frequency noise, as shown in Figure 5(D4). The wavelet transform [33,34] offers advantages in denoising, as it can effectively remove noise based on the distribution of signals and noise at different frequencies. This method is capable of efficiently representing the non-stationary characteristics, breakpoints, and abrupt changes in the signal. The wavelet transform of f(x) is shown as Equation (2):(2)$$T(a,b)=\frac{1}{a}\int_{R}f(t)\varphi \left(\frac{t-b}{a}\right)dt$$
where a is the scale factor that controls the scaling of the basic wavelet function φ(t), and b reflects the displacement. The shape of the Daubechies wavelet family resembles the QRS complex in ECG signals, and their energy spectrum is concentrated in the low-frequency range, making them suitable for ECG signal processing. The db5 wavelet from the Daubechies [35] wavelet family is used to decompose the signal, and can effectively remove high-frequency noise and baseline drift from the ECG signal. At each scale, the signal is decomposed into approximation components (A1–A5) containing the low-frequency components of the signal and detail components (D1–D5) containing high-frequency noise. The decomposition is shown in Figure 5.

In this study, an adaptive wavelet denoising algorithm [36,37] is employed to eliminate noise interference. The main frequency components of the ECG signal are located between 0.5 and 40 Hz, with a sampling frequency of $f_{s}=500\;\mathrm{hz}$ and $f_{L}=0.5\;\mathrm{hz}$. The db5 wavelet is used to decompose the signals into N = 10 levels. The calculation formula is Equation (3), and after thresholding, the signals are reconstructed to achieve noise removal. The choice of threshold significantly affects the denoising performance of the algorithm. For this purpose, the soft thresholding function [38] is utilized, with a threshold value denoted as $T=\delta \sqrt{2\mathrm{logN}}$. Noise below this threshold is effectively filtered out. Here, δ represents the calculated noise standard deviation based on Equation (4), and $d_{j}(k)$ denotes the k-th wavelet coefficient of the j-th level of the wavelet decomposition.
(3)$$N=\log_{2}(\frac{f_{s}}{f_{L}})$$
(4)$$\delta =\frac{median(d_{j}(k))}{0.6745}$$

The soft thresholding function sets the wavelet coefficients to 0 if their absolute values are below the threshold and scales them if they exceed the threshold. The calculation is performed as shown in Equation (5).
(5)$$W_{new}=\{\begin{array}{c} \mathrm{sgn}(w)(\left|W\right|-T)\\ 0 \end{array}\begin{array}{c} ,\\ , \end{array}\begin{array}{c} \left|W\right|\geq T\\ \left|W\right|<T \end{array}$$

The comparison between the raw signal and the results of the wavelet filtering effect is shown in Figure 6.

### 2.3. Recurrence Plot

The RP [39,40,41,42] is an important technique used to study the periodicity, chaotic nature, and non-stationary characteristics of time series, primarily for visual analysis in dynamic systems. The texture features of the RP represent the temporal information of the sequence and reflect the stationarity of the time series. The more stationary the time series is, the more uniform the distribution of the RP’s texture will be. Conversely, if the time series is non-stationary, the distribution of the RP’s texture becomes chaotic. Therefore, exploiting the characteristics of the RP can help uncover some latent information related to the onset of AF.

Although deep learning methods (1D CNN, RNN, LSTM, etc.) are capable of directly analyzing 1-D data, current research has primarily focused on processing 2-D structured data, particularly in the fields of computer vision (CV) and natural language processing (NLP) [43]. Therefore, by transforming 1-D time series into a 2-D recurrence matrix through a recursive relationship, the accuracy of model recognition can be improved. The following formula derivation will provide the dimensionality transformation of the data.

The key to the RP lies in obtaining the recurrence matrix. For a given time series x=x(i),x≤i≤N with length N, where N represents the length of the time series, we have an input signal consisting of 5000 data points; thus, N = 5000. The m-dimensional state space representation of the signal is Equation (6).
(6)$$u_{m}(i)=\{x(i),x(i+\tau ),\ldots ,x(i+(m-1)\tau )\}$$

In Equation (5), 1≤i≤N−(m−1)τ, and τ represents the time delay parameter, while m represents the embedding dimension parameter. In this study, based on experience, set τ is 3, and *m* is 3. The distance between any two vectors in the phase space is defined as Equation (7).
(7)$$Ri,j=\Theta (\varepsilon -||u_{m}(i)-u_{m}(j)||)$$
(8)$$\Theta (x)=\{\begin{array}{c} 1,x>0\\ 0,x\leq 0 \end{array}$$

In the given context, i,j=1,2,…,N−(m−1)τ,ε represent the predefined threshold, which is chosen as 10% of the maximum distance between any two vectors in the phase space. ||•|| represents the Euclidean norm, and Θ(x) is the Heaviside function defined by Equation (8).

It identifies a field centered on the vector $u_{m}(i)$ with ε as the radius. If the vector $u_{m}(j)$ is within this region, the time series is considered to exhibit recurrence and Ri,j=1; otherwise, Ri,j=0.

By applying the formula, the distance matrix ([N−(m−1)τ]×[N−(m−1)τ]) is transformed into a 4996 × 4996 binary matrix, ranging from 0 to 1. From the perspectives of point density, diagonal patterns, and horizontal and vertical structures, a set of quantitative evaluation metrics can be obtained. The RPs obtained from different types of ECG data are shown in Figure 7.

The Recurrence rate (REC) is the percentage of recurrent points, and is defined as Equation (9), where M is the dimension of the recurrence matrix.
(9)$$REC=\frac{1}{M^{2}}\sum_{i,j=1}^{M}R_{i,j}$$

Determinism (DET) measures the percentage of determinism, which quantifies the proportion of recurrence points that form diagonal structures in the recurrence plot; here $l_{\min}$ is 2, and is defined as Equation (10).
(10)$$DET=\frac{\sum_{l=l_{\min}}^{M}lp\left(l\right)}{\sum_{i,j=1}^{M}R_{i,j}}$$

RP Entropy (ENTR) is used to quantify the degree of disorder or complexity in the RPS, and is defined as Equation (11).
(11)$$ENTR=\sum_{l=l_{\min}}^{M}p\left(l\right)lnp\left(l\right)$$

Trapping time (TT) refers to the duration or time interval that a system’s trajectory remains in close proximity to a certain state before moving away, and is defined as Equation (12).
(12)$$TT=\frac{\sum_{v=v_{\min}}^{M}vp\left(v\right)}{\sum_{v=v_{\min}}^{M}p\left(v\right)}$$

Laminarity (LAM) is diagonal lines or clusters of points that indicate periods during which the system’s trajectory remains relatively constant or laminar, and is defined as Equation (13).
(13)$$LAM=\frac{\sum_{v=v_{\min}}^{M}vp\left(v\right)}{\sum_{i,j=1}^{M}R_{i,j}}$$

$L_{mean}$ measures the average length of the diagonal lines whose lengths exceed the certain threshold $l_{\min}$, and is defined as Equation (14).
(14)$$L_{mean}=\frac{\sum_{l=l_{\min}}^{M}lp\left(l\right)}{\sum_{l=l_{\min}}^{M}p\left(l\right)}$$

### 2.4. Residual Network

Convolutional Neural Networks (CNNs) [44] are a type of feedforward neural network with a grid-like hierarchical structure, capable of automatically extracting features without the need for manual intervention or prior expert knowledge. However, as the number of layers in the CNN increases, two significant issues have emerged: gradient vanishing, which affects network convergence, and the model’s accuracy tending to saturate. To address these issues, the block of a Residual Network (ResNet) [45,46] with shortcut connections is introduced. When the input is *x*, the target output is F(x)+x, thus avoiding the problem of performance degradation caused by an excessive number of convolutional layers.

The ResNet-based prediction network structure is shown in Figure 8, comprising three main components: CNN layers, improved ResBlocks, and fully connected layers. The first part is mainly used to process the input data and extract information such as the Recurrence rate, entropy, trapping time, and Lmax of the RP, which is then prepared for the subsequent deeper layers of the network. The second part utilizes the ResBlocks [47,48,49], which primarily consist of the following components: (1) Convolutional Layers: comprising two consecutive convolutional layers, each followed by a Batch Normalization layer and a non-linear activation function (ReLU). (2) Skip Connection: directly adds the input to the output of the convolutional layers. (3) Output: After the addition via the skip connection, the result y=F(x)+x is processed through a ReLU. The core of a ResBlock is the residual connection, which allows signals to bypass one or more layers in the network and propagate directly. If the two convolutional layers within the block do not learn any effective features, the signal can pass directly through the residual connection, achieving identity mapping. This means that the network can maintain its performance during training without degradation. Residual connections facilitate the direct flow of gradients to shallower layers, alleviating the vanishing gradient problem. The improved ResBlocks primarily employ local connections and parameter sharing in CNNs to reduce the number of model parameters. In ResBlock1, we use 2 × 2 convolutional kernels, while ResBlock2 and ResBlock3 utilize 3 × 3 convolutional kernels. By using multiple smaller convolutional kernels, each kernel interacts only with a local region of the input feature map, thus achieving local connectivity. The convolutional kernels slide over the input feature map to generate the output feature map. Each kernel applies the same set of weights at every position, thereby implementing parameter sharing. These two strategies help in reducing the number of model parameters. In the third part, there are two fully connected layers that map the output data features into a 1-D vector and produce the prediction results.

## 3. Results

### 3.1. Experimental Environment and Performance Indicators

The experimental environment used in this work is conducted on the PyCharm platform using a computer hardware configuration that includes an Intel(R) Core (TM) i7-6850 K CPU, an NVIDIA GeForce GTX 1080 Ti GPU, and 32 GB of RAM. The computer operating system is Windows 10, and the programming environment includes Python 3.7 and the open-source ML framework PyTorch [50]. The cross-entropy is used as the loss function for the network, and an Adam [51] optimization strategy with a learning rate of 10^−5^ is employed. The results obtained are all based on five-fold cross-validation.

In data analysis, precision, recall, F1 score, accuracy, and AUC are calculated as metrics to compare the performance of different models. The evaluation metrics are defined as Equations (15)–(18).
(15)$$Accuracy=\frac{TP+TN}{TP+TN+FP+FN}$$
(16)$$Precision=\frac{TP}{TP+FP}$$
(17)$$Recall=\frac{TP}{TP+FN}$$
(18)$$F1=\frac{2TP}{2TP+FP+FN}$$
where TP is the number of correctly classified positive samples, TN is the number of correctly classified negative samples, FN is the number of positive samples incorrectly classified as negative samples, and FP is the number of negative samples incorrectly classified as positive samples. Recall reflects the model’s ability to correctly identify true AF samples. Improving recall effectively reduces the risk of missing detections of AF patients, ensuring their safety.

### 3.2. Performance

To evaluate our proposed method, the four features with statistically significant discrimination are extracted from the presented dataset, including the Recurrence rate, entropy, trapping time, and Lmax, shown in Table 2. The Recurrence rate is higher in the Pre-AF group as compared to the Non-AF group. It can be related to larger dynamic periodicity in the episodes before AF events. The increased entropy values in Pre-AF are associated with greater complexity in the RP structure before AF episodes. Compared to Non-AF episodes, the trapping time, measuring the average duration of vertical structures, shows higher values in Pre-AF. This implies that the system maintains a specific state for a longer average duration before AF compared to Non-AF. Lmax also exhibits higher values in Pre-AF; this can be interpreted as the stable interaction time before AF being longer compared to Non-AF. These features indicate that the RP structure contains valuable information characterizing different levels of chaos between AF signals with different types. This will contribute to neural networks better extracting implicit information preceding AF, thereby improving prediction accuracy.

The loss values and accuracy curves of the AFPDB dataset are shown in Figure 9. The training performance (red curve) is better than the testing (blue curve). After 38 epochs, overfitting is observed, indicating the need to implement early stopping. The best results are achieved after 38 epochs of training. Similarly, the loss and accuracy curves of the model on the presented dataset are shown in Figure 10. After 175 epochs, overfitting is observed, indicating the need to implement early stopping. The best results are attained after 175 epochs of training.

This paper conducts experiments based on two datasets and compares various prediction models. Evaluation metrics such as accuracy are utilized to identify the optimal model for early diagnosis of AF in patients. As shown in Table 3 and Table 4, the experimental results are obtained for different models. Since the improved ResBlock is used for model building, a comparative experiment is conducted using Resnet18. The performance metrics are close to our proposed model, but the precision is lower than that of our proposed model. In DenseNet, each layer is directly connected to all preceding layers, enabling the reuse of features and reducing the number of model parameters. Compared to ResNet, DenseNet addresses the issues of low feature utilization and the large parameter count in ResNet. In this experiment, the parameter size of DenseNet is 27.1 MB. However, in our improved ResBlock, we utilize a CNN with local connections and parameter sharing to reduce the number of model parameters. The parameter size of our model is only 10.8 MB, and it outperforms DenseNet in terms of model performance. The proposed model-1D takes the filtered ECG signal as input directly into the model for training without converting it into an RP. It performs worse than the proposed model-2D in all the evaluation metrics. The confusion matrices of different models on the AFPDB dataset are shown in Figure 11a–d. Our proposed model achieves a prediction accuracy of 97.0%, which is higher than other models. The diagonal line in the middle of the ROC curve is referred to as the “random chance line”. Models located above this line are considered to perform better than random chance at certain thresholds. The ROC curves of different models on the AFPDB dataset are shown in Figure 12, AUC values of 91.5%, 91.8%, 90.9%, and 99.7% for four models. Our proposed model outperforms the other comparison models as well. The confusion matrices of different models on the dataset from the presented dataset are shown in Figure 11e–h. Our proposed model achieves a prediction accuracy of 93.4%, which is higher than other models. The ROC curves of different models on the presented dataset are shown in Figure 13, AUC values of 95.9%, 90.8%, 89.7%, and 96.0% for four models. Our proposed model also outperforms the other comparison models. The improvement in model metrics has two significant benefits: Increasing the recall rate can reduce missed diagnoses, which is crucial for the early diagnosis and treatment of diseases. Enhancing trust in the monitoring system, as medical professionals are more likely to trust and rely on the predictive system as an auxiliary diagnostic tool when they see it can reliably identify AF cases.

## 4. Discussion

The comparative results of the latest research on the same dataset are shown in Table 5. YANG Ping et al. [52] proposed a hybrid model combining CNN and LSTM (CNN-LSTM) to extract the local spatial features and temporal dependency features embedded in pattern transition features, and achieved an accuracy of 91.3% and a recall of 82.2%. The model exhibited a low recall, indicating a weaker ability to detect positive samples. Further improvement is needed in this aspect. Liang-Hung Wang et al. [53] focused on second-lead ECG signals and extracted eight temporal and six frequency domain features for AF prediction. An improved quantum particle swarm optimization algorithm (IQPSO) combined with SVM was utilized to construct an effective predictive model for AF, and achieved a recall of 94.2% and an accuracy of 87.0%. However, it required manual intervention and did not achieve an end-to-end approach. Cai Wu et al. [23] simultaneously extracted P-wave morphological parameters and heart rate variability features from ECG, and achieved an accuracy of 92.5%, a recall of 88.0%, and an F1 score of 92.3%. The model demonstrated superior performance in machine learning models, but it also did not achieve an end-to-end approach. Le Sun et al. [24] developed a recursive neural network called SLAP, which consists of stacked LSTM layers, for AF prediction, and achieved an accuracy of 92.0%, and an F1 score of 92.0%. On the AFPDB dataset, it outperformed other models in terms of these metrics. Compared to machine learning algorithms, the end-to-end approach of implementing deep learning algorithms demonstrates superior performance, especially at large data scales. The improvement in model metrics is primarily due to the application of the following methods: wavelet filtering can remove noise interference, utilizing the features of RP can help uncover potential information related to AF episodes, while the improved ResNet network excels at recognizing varying degrees of chaos among different types of AF signals. Innovatively combining RP with improved ResNet maximizes the extraction of hidden information from ECG signals, thereby enhancing the performance of AF prediction. Clinical application: Early-onset AF is characterized by its sudden, intermittent, and short-lasting episodes, which are often not easily captured by clinical ECG examinations. Many patients with suspected AF visit the clinic due to recurrent chest discomfort and palpitations during the inter-episode period, at which time the ECG appears normal, leading to a high rate of missed diagnoses for clinical AF. Moreover, the use of delayed monitoring devices to detect AF episodes is cost-inefficient, and widespread implementation could lead to a waste of healthcare resources. Currently, our model has been used for auxiliary diagnosis at the Fourth Affiliated Hospital of Zhejiang University. The use of this model provides the benefits of offering early warnings to patients, reducing the risk of complications caused by AF, optimizing the allocation of medical resources (if the model predicts a risk of developing AF, then delayed monitoring devices are required for further monitoring; otherwise, they are not necessary), and reducing patient costs. Limitations: Currently, the model is only capable of preliminary screening and cannot perform real-time monitoring for patients. Our next step is to consider how to optimize the model, reduce the number of model parameters, and deploy it to wearable devices to achieve real-time monitoring capabilities [54,55,56].

## 5. Conclusions

In this research, we present a prediction model that combines RP and ResNet architecture to optimize the accuracy of predicting AF. The model incorporates a wavelet filtering algorithm to minimize noise interference, ensuring robust prediction results. By leveraging the recursive relationship, we generate a two-dimensional graph and derive a set of quantitative indicators. Ultimately, these indicators are utilized by our proposed model for AF prediction. During evaluation, the proposed model achieved an outstanding accuracy of 97.0%, outperforming existing methods. This significant improvement underscores the benefits of integrating wavelet denoising and RP techniques in AF prediction. On our proprietary dataset, the model demonstrated strong performance, with a prediction precision, recall rate, F1 score, accuracy, and AUC of 90.5%, 89.1%, 89.8%, 93.4%, and 96%, respectively. These results underscore the practical utility of our approach. For future work, there are several avenues for future research. One possibility is to examine a wider range of abnormal slices to enhance the model’s ability to generalize. Additionally, expanding the dataset through more extensive patient studies can further enrich the model’s predictive capabilities. Another promising direction is to integrate the AF prediction model into intelligent wearable devices for real-time monitoring and early warning systems, potentially revolutionizing AF management and treatment.

## Figures and Tables

**Figure 1 sensors-24-04978-f001:**
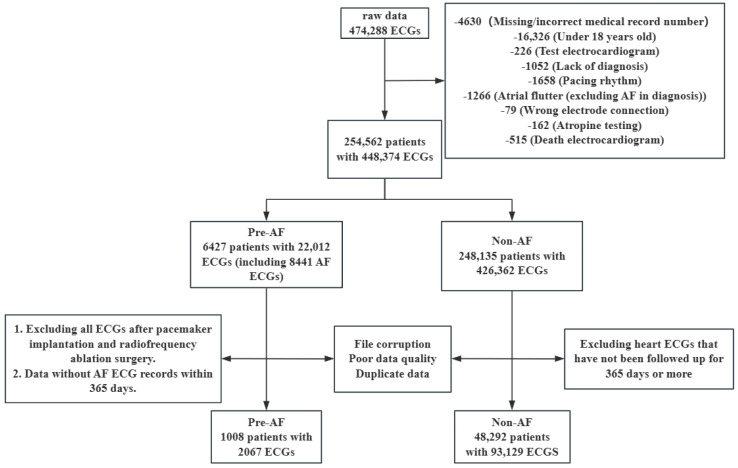
Flowchart for the screening and grouping process of routine 12-lead ECGs.

**Figure 2 sensors-24-04978-f002:**
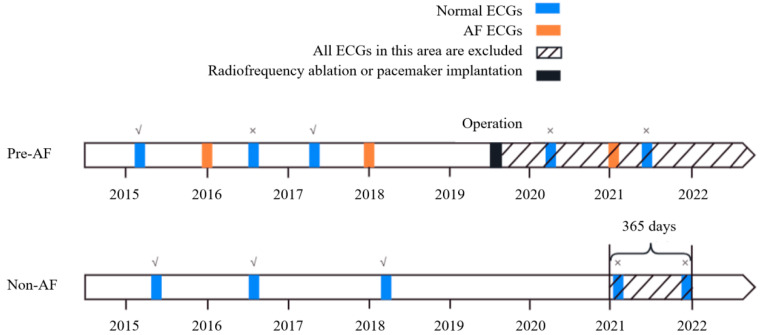
Schematic diagram of ECG selection for the Pre-AF group and Non-AF group (The ECGs inclusion and exclusion processes for both groups are shown, respectively. “√” represents inclusion, “×” represents exclusion).

**Figure 3 sensors-24-04978-f003:**
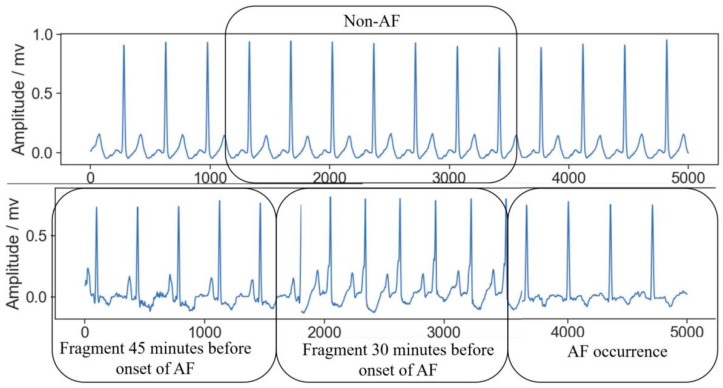
Relationship between different types of records.

**Figure 4 sensors-24-04978-f004:**
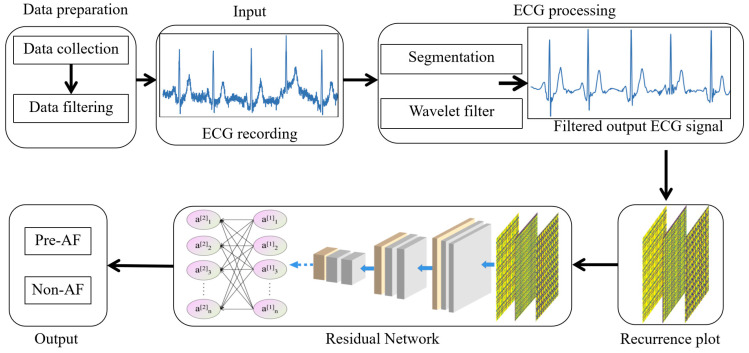
Data processing flowchart.

**Figure 5 sensors-24-04978-f005:**
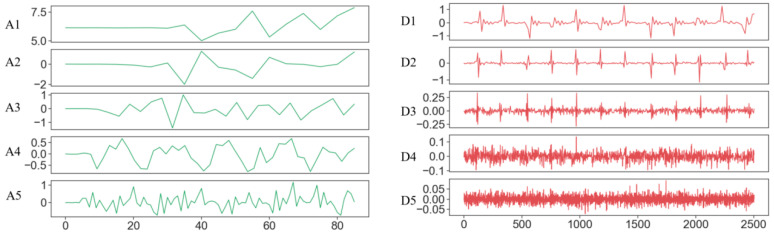
Wavelet decomposition schematic diagram (*x*-axis represents the number of data points, *y*-axis represents the amplitude/mv).

**Figure 6 sensors-24-04978-f006:**
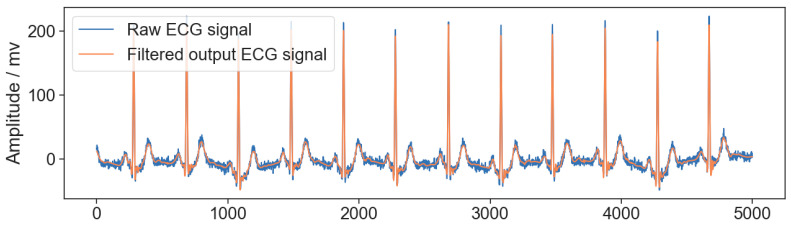
Comparison of the raw ECG signal and wavelet filtering.

**Figure 7 sensors-24-04978-f007:**
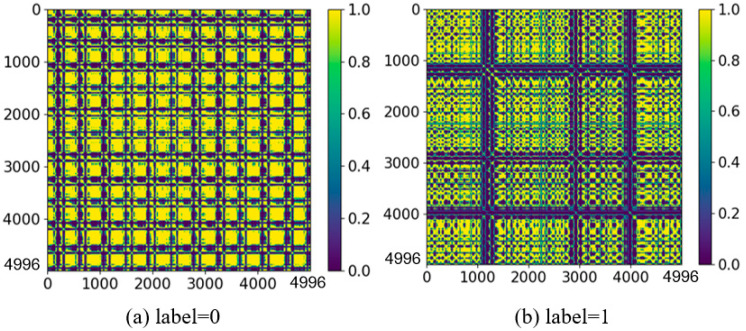
RPs generated by different ECGs (The *x*-axis and *y*-axis represent the number of data points).

**Figure 8 sensors-24-04978-f008:**
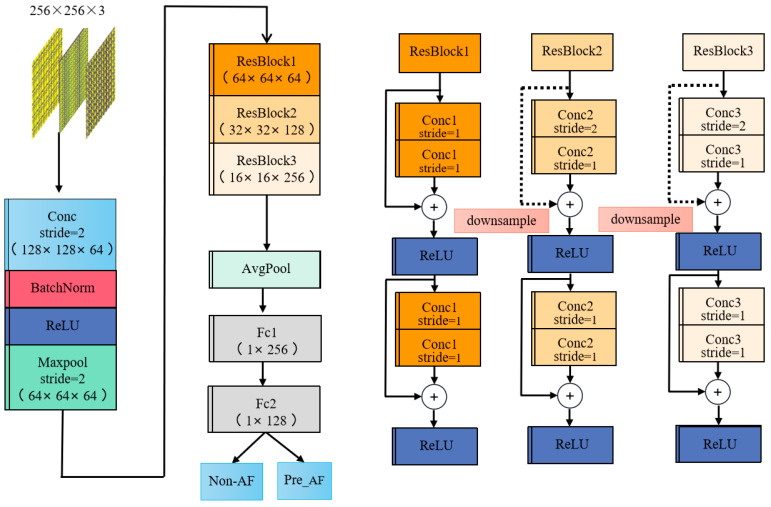
Prediction model structure diagram (the structure diagram is the proposed model-2D).

**Figure 9 sensors-24-04978-f009:**
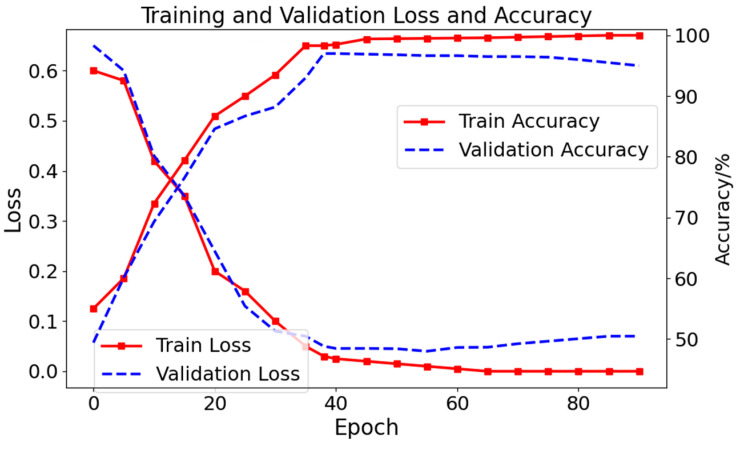
Loss values and accuracy of the model on the AFPDB dataset (overfitting is observed after 38 epochs).

**Figure 10 sensors-24-04978-f010:**
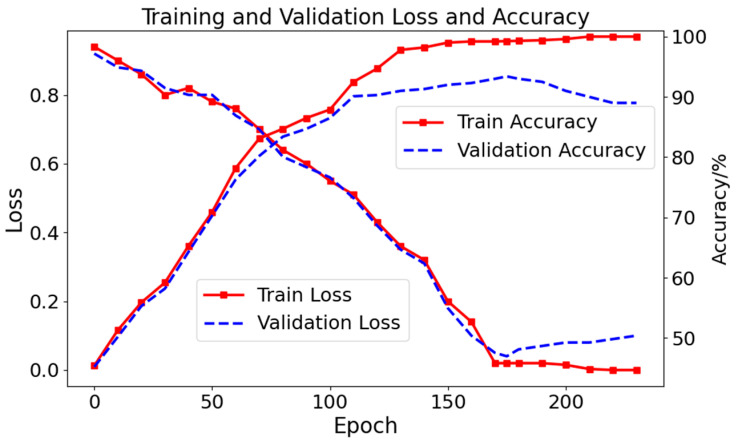
Loss values and accuracy of the model on the presented dataset (overfitting is observed after 175 epochs).

**Figure 11 sensors-24-04978-f011:**
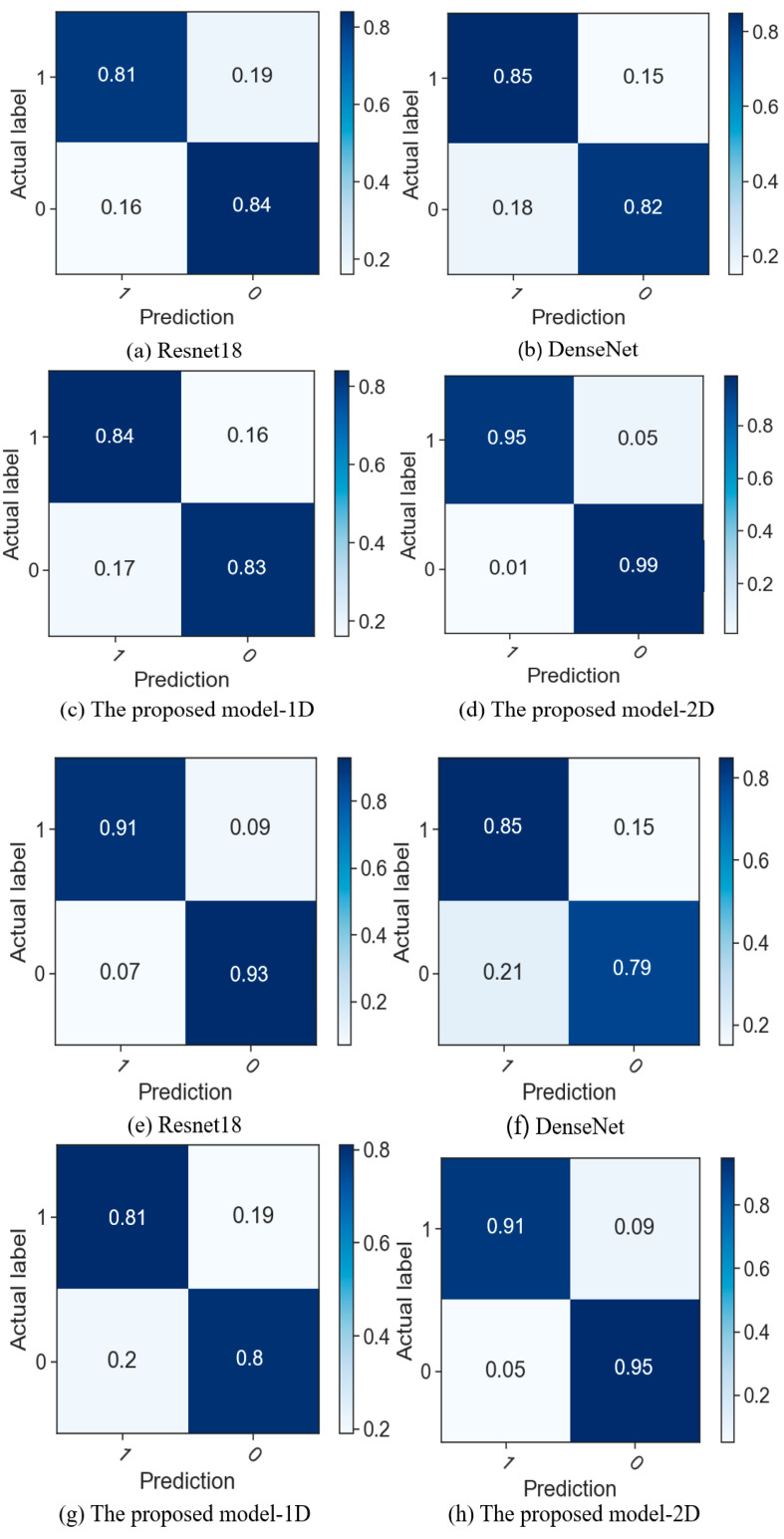
(**a**–**d**) are confusion matrices of four models on AFPDB dataset; (**e**–**h**) are confusion matrices of four models on the presented dataset.

**Figure 12 sensors-24-04978-f012:**
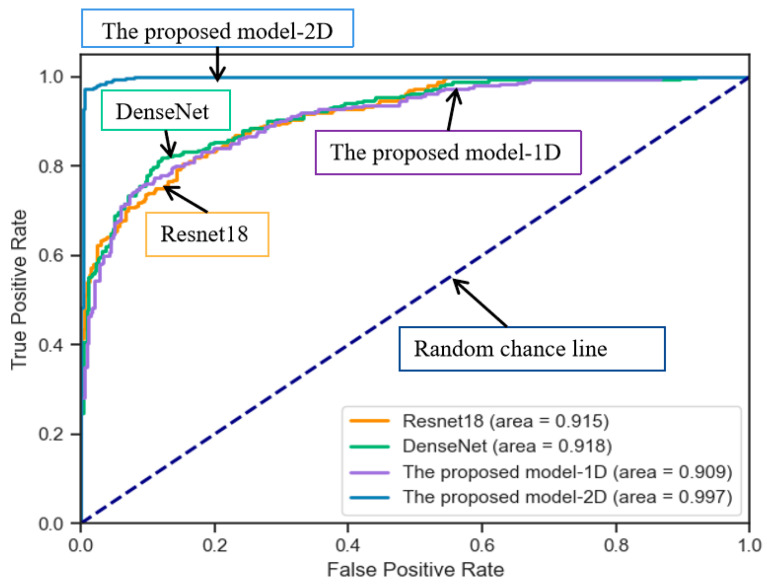
ROCs of four models on AFPDB dataset (the proposed model-2D obtains the best results, AUC = 0.997).

**Figure 13 sensors-24-04978-f013:**
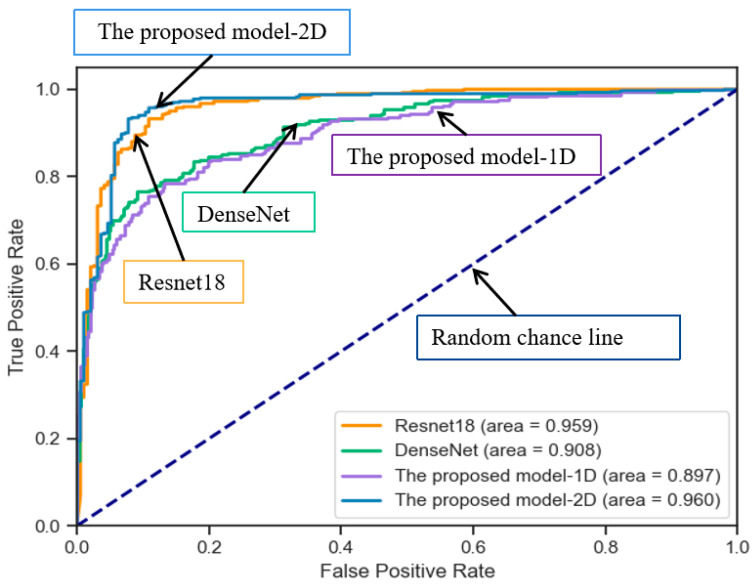
ROCs of four models on the presented dataset (the proposed model-2D obtains the best results, AUC = 0.96).

**Table 1 sensors-24-04978-t001:** General information of Pre-AF group and Non-AF group.

Groups	Pre-AF Group (*n* = 2067)	Non-AF Group (*n* = 93,129)
Age (years)	70.5 ± 13.1	49.6 ± 16.0
Gender (male)	1324 (64.1%)	48,530 (52.1%)

**Table 2 sensors-24-04978-t002:** Mean value and standard deviation of the features for Pre-AF and Non-AF (Pre-AF is one month prior to the onset of AF).

	Mean ± STD		
Feature	Pre-AF	Non-AF	*p*-Value
Recurrence rate	0.544 ± 0.182	0.479 ± 0.127	0.00095
Determinism	0.999 ± 0.033	0.998 ± 0.035	0.814
Entropy	0.462 ± 0.169	0.441 ± 0.141	0.042
Trapping time	0.726 ± 0.028	0.605 ± 0.023	0.00037
Laminarity	0.999 ± 0.016	0.997 ± 0.015	0.712
Lmax	0.369 ± 0.125	0.324 ± 0.129	0.008
Lmean	0.172 ± 0.136	0.166 ± 0.128	0.078

**Table 3 sensors-24-04978-t003:** Comparison of four models’ prediction results on the AFPDB dataset.

Models	Precision/%	Recall/%	F1 Score/%	Accuracy/%
Resnet18	80.5	85.2	82.8	82.3
DenseNet	85.4	84.9	85.2	83.9
The proposed model-1D	85.4	84.4	84.9	83.6
The proposed model-2D	94.8	99.4	97.0	97.0

**Table 4 sensors-24-04978-t004:** Comparison of four models’ prediction results on the presented dataset.

Models	Precision/%	Recall/%	F1 Score/%	Accuracy/%
Resnet18	90.5	83.9	87.1	91.9
DenseNet	85.4	82.6	84.0	82.4
The proposed model-1D	81.5	82.6	82.0	80.7
The proposed model-2D	90.5	89.1	89.8	93.4

**Table 5 sensors-24-04978-t005:** Comparison of related work under the AFPDB dataset.

Methods	Precision/%	Recall/%	F1 Score/%	Accuracy/%
Method1 [52]	91.1 (calculate)	82. 2	86.4 (calculate)	91.3
Method2 [53]	92.8 (calculate)	93.3	93.0 (calculate)	92.5
Method3 [23]	97.0 (calculate)	88.0	92.3	92.0
Method4 [24]	92.0 (calculate)	92.0	92.0	92.0
Proposed method	94.8 (+2%)	99.4 (+6.1%)	97.0 (+4%)	97.0 (+4.5%)

## Data Availability

Data are contained within the article.

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
