# Peer review of "Atrial Fibrillation Prediction Based on Recurrence Plot and ResNet"

_sensors, 2024, doi:10.3390/s24154978_

Round 1

Reviewer 1 Report

Comments and Suggestions for Authors

Review of Manuscript: "Atrial Fibrillation Prediction Based on Recurrence Plot and ResNet"

General Assessment

This manuscript presents a method combining the Recurrence Plot (RP) technique and the ResNet architecture to predict atrial fibrillation (AF) based on ECG data. The topic is highly relevant and presents a novel approach to AF prediction, with promising results. Below is a detailed evaluation based on the Kufa Medical Journal (KMJ) reviewer guidelines.

Novelty and Originality

The manuscript introduces an innovative combination of RP and ResNet architecture for AF prediction, showcasing a high level of novelty and originality. The method's ability to achieve high prediction accuracy and generalizability across different datasets highlights its contribution to the field.

Scientific Reliability

The methodology is well-detailed, with robust data preprocessing steps and comprehensive evaluation metrics. The manuscript provides a clear explanation of the experimental setup, including the databases used, data processing techniques, and model architecture. The results are validated through extensive experiments, demonstrating high levels of precision, recall, F1 score, and AUC.

Valuable Contribution to Science

The proposed method significantly enhances the prediction of AF, particularly in early stages, which is critical for timely medical intervention. The high performance metrics on both proprietary and public datasets underscore the practical applicability of the approach.

Adding New Aspects to the Field

The manuscript adds value by integrating wavelet denoising and RP techniques with deep learning, which is relatively unexplored in AF prediction. The detailed analysis of RP features and their impact on prediction performance offers new insights into ECG signal processing.

Ethical Aspects

The manuscript includes an ethics approval statement and informed consent from participants, ensuring compliance with ethical standards. There are no apparent ethical concerns regarding the study's conduct.

Structure and Relevance to Author Guidelines

The manuscript is well-structured, following a logical flow from introduction to conclusion. Each section is clearly defined, and the content is relevant to the study's objectives. The inclusion of figures and tables enhances the clarity of the presented information.

References

The references are comprehensive and up-to-date, supporting the manuscript's claims. However, a few additional citations to recent works in deep learning applications for AF prediction could further strengthen the literature review.

Grammar, Punctuation, and Spelling

The manuscript is well-written, with minor grammatical and typographical errors. A thorough proofreading is recommended to ensure clarity and professionalism.

Scientific Misconduct

No evidence of scientific misconduct was found in the manuscript. The authors have appropriately cited their sources, and there is no indication of plagiarism.

Specific Comments

  1. Abstract:

    • The abstract effectively summarizes the study but could be more concise. Consider reducing redundancy to highlight the key findings and contributions more clearly.
  2. Introduction:

    • Provide more context on the prevalence and impact of AF. Discuss why combining RP with ResNet is a novel and effective approach.
  3. Methodology:

    • The methodology section is comprehensive. However, a flowchart summarizing the entire process from data collection to prediction could enhance understanding.
  4. Results:

    • The results are well-presented with appropriate use of tables and figures. Consider adding a discussion on the implications of the findings for clinical practice.
  5. Discussion and Conclusion:

    • The discussion should include potential limitations of the study and suggestions for future research. The conclusion effectively summarizes the study's contributions and practical implications.
  6. Figures and Tables:

    • Ensure that all figures and tables are referenced in the text. Figure legends should be concise yet descriptive enough to stand alone.

Conclusion

The manuscript presents a highly relevant and novel approach to AF prediction, demonstrating significant improvements over existing methods. With minor revisions to enhance clarity and address potential limitations, the manuscript will make a valuable contribution to the field.

Recommendation: Accept with Minor Revisions

Reviewer 2 Report

Comments and Suggestions for Authors

In this paper, a prediction model combining RP and ResNet architectures is proposed to optimise the accuracy of AF prediction. The model incorporates a wavelet filtering algorithm to minimise noise interference and achieves better results. It provides ideas for AF prediction. However, the article still has some problems to be discussed:

1.      The novelty of the paper should be expressed in the main text. Especially, the comparison parts between the proposed scheme and the existing results should be given.

2.      In this paper 2.2 the reason for the choice of denoising threshold is not described clearly enough, whether this threshold is better for denoising. Please provide an explanation.

3.      To what extent does the selection of parameter values affect the results and performance of the experiment? I am curious about how to determine whether the improvement in experimental results and performance is due to the innovation of the algorithm itself, rather than the influence of the parameter values. Please describe it in detail.

4.      In this paper 2.4 the principle of ResBlock is not described clearly enough.

5.      In this paper 3, it is mentioned that in the improved ResBlock, CNNs with local connectivity and parameter sharing are used to reduce the number of model parameters, please specify how this is done.

6.      It is necessary to prove that the proposed method can guarantee the desired system performance. Please give the further presentation.

Reviewer 3 Report

Comments and Suggestions for Authors

The manuscript presents a novel approach combining the Recurrence Plot (RP) technique and the ResNet architecture to predict atrial fibrillation (AF) from electrocardiogram (ECG) recordings. The method shows promising results, with high prediction precision, recall, F1 score, accuracy, and AUC both on a proprietary dataset and a publicly available dataset (AFPDB). However, several critical issues need to be addressed before the manuscript can be considered for publication. 1.The methodology section should provide sufficient information for replication. Currently, key parameters, settings, and the rationale behind the choices made in the methodology are missing. 2. Fig. 3 and 4 are hard to follow by readers, please check. 3. The experimental results should be accompanied by statistical significance tests to validate the performance improvements claimed. What are the advantages compared to current advanced methods? 4. Eq 8-11 seems to be well-known, which are no need to be claimed here. 5. From Figure 13, it seems that the improvement of the proposed method is not very sufficient.

Comments on the Quality of English Language

The presence of a large number of abbreviations in the paper reduces its readability, please check.

Round 2

Reviewer 3 Report

Comments and Suggestions for Authors

I have no further comments.